# GM1 Ganglioside Is A Key Factor in Maintaining the Mammalian Neuronal Functions Avoiding Neurodegeneration

**DOI:** 10.3390/ijms21030868

**Published:** 2020-01-29

**Authors:** Elena Chiricozzi, Giulia Lunghi, Erika Di Biase, Maria Fazzari, Sandro Sonnino, Laura Mauri

**Affiliations:** Department of Medical Biotechnology and Translational Medicine, University of Milano, 20090 Segrate, Milano, Italy; elena.chiricozzi@unimi.it (E.C.);

**Keywords:** GM1 ganglioside, GM1 neuronal function, GM1 in neuronal development, GM1 in neurodegeneration, GM1 oligosaccharide function

## Abstract

Many species of ganglioside GM1, differing for the sialic acid and ceramide content, have been characterized and their physico-chemical properties have been studied in detail since 1963. Scientists were immediately attracted to the GM1 molecule and have carried on an ever-increasing number of studies to understand its binding properties and its neurotrophic and neuroprotective role. GM1 displays a well balanced amphiphilic behavior that allows to establish strong both hydrophobic and hydrophilic interactions. The peculiar structure of GM1 reduces the fluidity of the plasma membrane which implies a retention and enrichment of the ganglioside in specific membrane domains called lipid rafts. The dynamism of the GM1 oligosaccharide head allows it to assume different conformations and, in this way, to interact through hydrogen or ionic bonds with a wide range of membrane receptors as well as with extracellular ligands. After more than 60 years of studies, it is a milestone that GM1 is one of the main actors in determining the neuronal functions that allows humans to have an intellectual life. The progressive reduction of its biosynthesis along the lifespan is being considered as one of the causes underlying neuronal loss in aged people and severe neuronal decline in neurodegenerative diseases. In this review, we report on the main knowledge on ganglioside GM1, with an emphasis on the recent discoveries about its bioactive component.

## 1. Introduction

Any scientist doing research in the field of gangliosides, sialic acid containing glycosphingolipids eventually encounters the ganglioside GM1. GM1 is surely the most used ganglioside in an incredible variety of biological experiments aimed at understanding the physiological role played by gangliosides. Knowledge of GM1 dates back to 70 years ago, with its structural characterization established in 1963 [1,2]. Several aspects probably contributed to its popularity: it is abundant in all mammalian brains, where it covers 10%–20% of the total ganglioside mixture [3,4]; it has been described as the intestinal receptor for cholera toxin [5]; it was rapidly proposed as a neurotrophic and neuroprotective compound [2,6] therapeutically used for diabetic and peripheral neuropathies [7,8] so that several clinical studies were conducted with the aim to test GM1 efficacy in a broad range of neurological diseases [7].

Today, thanks to the myriad of studies and data reported in the literature, ever increasing information on how GM1 modulates some protein activity and the downstream cell signaling, acting on the cell biology, is becoming available.

In this review, we report the state of art of the knowledge on ganglioside GM1.

## 2. Structure of Human Brain Gangliosides

Gangliosides are glycosphingolipids containing sialic acid (Figure 1, A1 and A2). The human brain contains many gangliosides [9,10], but five of them cover altogether over 95%–97% of the total content, namely GM1 (Figure 1B), GD1a, GD1b, GT1b and GQ1b. This nomenclature was introduced by Lars Svennherolm in 1980 [11] and it is accepted by the IUPAC-IUB commission for nomenclature of lipids [12]. The structure of these gangliosides is reported in Table 1. They belong to the ganglio-4 series, Gg4, characterized by the neutral oligosaccharide β-Gal-(1-3)-β-GalNAc-(1-4)-β-Gal-(1-4)-β-Glc-.

The structure of GM1 was established in 1963 [1], resulting in humans to have β-Gal-(1-3)-β-GalNAc-(1-4)-[α-Neu5Ac-(2-3)-]β-Gal-(1-4)-β-Glc-(1-1)-Cer. Structurally, sialic acid linked to gangliosides can be N-acetylneuraminic acid, Neu5Ac, N-glycolylneuraminic acid, Neu5Gc, as well as O-acetylated sialic acid [13]. Nevertheless, a detailed study published in 1970 clearly demonstrated that humans do not contain Neu5Gc [14] and in the following years, it was shown that the gene coding for the CMP-Neu5Ac oxidase was deleted in humans [15].

In 1975, the structure of a different GM1 present into the brain, α-Neu5Ac-(2-3)-β-Gal-(1-3)-β-GalNAc-(1-4)-β-Gal-(1-4)-β-Glc-(1-1)-Cer was determined [16]. This one is now coded as GM1b, while sometimes, the GM1 with the sialic acid linked to the internal galactose is coded GM1a. In this paper, GM1 refers only to GM1a.

The lipid moiety of gangliosides is named ceramide, Cer, (Figure 1C), deriving from the Latin word “*cera*” (vax), due to its high hydrophobicity. In the brain, ceramide is constituted of long-chain amino alcohol, (2S,3R,4E)2-amino-1,3-dihydroxy-octadecene/eicosene, known as sphingosine C18- or C20-sphingosine (octadecene or eicosene, respectively) according to the number of carbon atoms, connected to a fatty acid by an amide linkage; C20-sphingosine is present quite exclusively in the brain [14,17]. A minor quantity of the saturated species, called sphinganine, is also present. In the brain, over 90% of the total acyl chains linked to the amino group of sphingosine is covered by stearic acid.

## 3. Gangliosides as the Main Actors in Membrane Organization: The Amphiphilic Properties of Gangliosides

The presence of an extended double carbon tail confers a strong hydrophobic character to gangliosides. On the other hand, the hydrophilic character is also highly expressed, due to the negative charge of sialic acid, which becomes progressively more pronounced as the complexity of the head group increases in an extended and branched structure by the addition of sialic acid residues.

Thus, the hydrophilic–hydrophobic balance is the competition between two opposite strong requirements and any variation in such requirements affects the amphiphilic character [18].

As amphiphilic compounds, gangliosides aggregate in solution, forming quite small spherical/ellipsoidal micelles. Micelles are in equilibrium with the monomers, with a critical micellar concentration (c.m.c.) in the range of 10^−5^ M–10^−9^ M. The value is determined by the amphiphilic character, i.e., by the ceramide structure (mainly by the length of acyl chain) and by the number of residues of sialic acid. This means that a scant number of monomers can be available in solutions, unless associated with soluble proteins [18,19,20]. Brain GM1 has a c.m.c. between 10^−8^ M–10^−9^ M [18], but this value increases of over three orders of magnitude reducing the length of the acyl chain from 18 to 2 carbons [18], confirming a specific role of the ceramide moiety in determining the physico-chemical properties of gangliosides, while maintaining constant the oligosaccharide chain.

In gangliosides, the double bond close to the head group promotes the parallel orientation of the axes of the two hydrocarbon chains [18,21]. This, together with the hydrophobic character of ceramide, allows gangliosides to be components of the membranes, being particularly abundant in plasma membranes [22]. Gangliosides are synthesized in the Golgi apparatus and reach the plasma membranes by vesiculation, then becoming components of the external membrane layer [23]. The ceramide moiety remains inserted into the layer, reducing the membrane fluidity [24]; while the glycan protrudes into the extracellular milieu, and thus is allowed to interact with neighboring membrane components (cis interactions), with membrane components of neighboring cells (trans interactions), and with soluble intercellular molecules and with water as well. It is important to consider that the huge branched chain of gangliosides associated with water strongly interacting with it, occupies a large volume, so that homogeneous distribution of gangliosides at the cell surface is thermodynamically disadvantaged. Thus, dynamic segregation phenomena involving gangliosides occur at the cell surface with the formation of membrane lipid domains attracting large amount of cholesterol, now worldwide known with the name “lipid rafts” [24,25]. Today, although improperly (see below), ganglioside GM1 is often considered as a marker of lipid rafts.

## 4. GM1 of Neuronal Plasma Membrane in the Central Nervous System: Changes of Its Content During Aging and in Neurodegenerative Diseases

As for all gangliosides, GM1 is synthesized on the luminal membrane of Golgi apparatus, becomes a component of released Golgi vesicles and finally associates to the outer layer of plasma membranes by vesicle–plasma membrane fusion [26,27]. In addition to the biosynthetic process, GM1 membrane content is determined by the total metabolic pathway involving intracellular trafficking, and lysosomal catabolism [26,27]. A portion of the membrane GM1 derives from the hydrolysis of polysialylated gangliosides, particularly of GD1a, by the membrane associated sialidase Neu3 [28,29]. Sialidase Neu3 expression is induced by specific signals, including the one determined by phosphatidylinositol that facilitates the transmembrane insertion of the enzyme [29]. Sialidase Neu3 works in *trans* manner [28], but a *cis* activity cannot be excluded.

Ganglioside content has been studied in detail in the human brain of subjects ranging between 20 and 100 years of age [10]. The brain sialic acid ganglioside amount progressively decreases, being reduced of about 30% in centenary in comparison with 20 year-old people. The ganglioside pattern undergoes changes during aging with an increased proportion of b-series gangliosides, particularly GD1b, and a reduced content of a-series gangliosides, including GM1 and GD1a; this latter can be considered as a reservoir for the production of GM1 at the plasma membrane. Thus, GM1 and GD1a are the two gangliosides mainly responsible for the decrease of sialic acid ganglioside content along the aging of humans.

Alterations of ganglioside pattern can be found in different neurodegenerative diseases, including Parkinson, Alzheimer and Huntington diseases. The ganglioside mean content in the brain from seven Parkinson’s disease male with age ranging from 67 to 95 was found to be reduced by 22% compared to the brain content of 12 healthy controls with the same age. No difference was found between control and Parkinson’s disease females [30].

The ganglioside content of specific brain areas from patients affected by Alzheimer’s disease was evaluated [9,31]. In total, in temporal cortex from 10 patients with a mean age of 83 years, reflecting late stages of Alzheimer’ pathology, the content of ganglioside is reduced by 45%. Nevertheless, in these subjects, the absolute content of gangliosides in lipid rafts is similar to that present in lipid rafts obtained from control subjects of the same age. In the frontal cortex of 10 patients of an average age of 72 years and presenting earlier stages of the pathology, the ganglioside content was comparable to that of a matched-aged control group, but the percentage of gangliosides in lipid rafts showed an increase, particularly for GM1, which doubled. A significantly higher GM1 content was also found in lipid rafts prepared from platelets purified from Alzheimer patients, in comparison with the control group [32].

There is no information on the brain ganglioside content in cases of human Huntington’s disease, but the ganglioside synthesis was shown to be decreased in cells and animal models of the disorder [33]. A similar result was observed in the fibroblasts from humans Huntington’s disease [9].

## 5. GM1 Preparations and GM1 Analysis

Pure ganglioside GM1, single species of GM1 with homogeneous ceramide moiety and ganglioside GM1 containing probes are necessary for biological studies.

### 5.1. Preparation of GM1 with Heterogeneous Ceramide Moiety

The main source of ganglioside GM1 is, still today, the mammalian brain: pig, sheep and calf are the most used animal sources. Total brain lipids are extracted with solvents and gangliosides separated by neutral glycosphingolipids and phospholipids by partitioning [34]. GM1 is purified by chromatographic procedures from the total ganglioside mixture [35,36,37,38,39,40]. 

The chemical synthesis of ganglioside GM1 has been reported [2], but it remains an approach that allows the production of small amounts of ganglioside and limited to specialized laboratories. Moreover, great care must be taken on the final purity, avoiding the presence of minor contaminants that could be toxic for cells and animals used in biological experiments. Any chromatographic procedure requires a solid support (silica gel, Iatrobeads, sepharose or sephadex of different size) and gangliosides are largely absorbed by these supports; thus, particular care is necessary to avoid low yields.

Ion exchange column chromatography fractionation is probably one of the most used and simple procedure for GM1 purification [38]. Gangliosides are fractionated by anion exchange column chromatography on the basis of their sialic acid content. Gangliosides solubilized in methanol are loaded on the column and eluted with ammonium or sodium acetate in methanol at progressively increasing concentration. The monosialoganglioside fraction, of which GM1 represents 90%–95%, is eluted at a low salt concentration. Minor gangliosides are then subsequently removed by silica gel chromatography [38].

When a small amount of ion exchange support is used, in comparison with the added ganglioside mixture, the polyanionic ganglioside mixture itself acts as a detaching system for GM1 [41]. Thus, the continuous addition of the ganglioside mixture to the ion exchange chromatography column exceeds the support binding capacity and GM1 is rapidly displaced by polysialogangliosides.

Pretreatment of the total ganglioside mixture with sialidase is useful to increase the content of ganglioside GM1 into the ganglioside mixture [42,43,44]. Sialidases, with a few exceptions, do not hydrolyze the α2-3 linkage connecting sialic acid to the internal galactose of the oligosaccharide chain because of steric hindrance exerted by N-acetylgalactosamine. Exhaustive sialidase treatment of the brain ganglioside mixture transforms the natural ganglioside mixture into GM1 together with minor quantities of GM2 (the endogenous GM2 and that deriving from GD2 by the enzyme activity) and GalNAc-GD1a [43,44].

A final column chromatography purification is necessary to have homogeneous GM1.

### 5.2. Preparation of GM1 with Homogenous Ceramide Moiety

GM1 prepared according to the above procedures is heterogeneous in the ceramide moiety. GM1 species with homogeneous C18- or C20-sphingosine, but still with some minor acyl chain heterogeneity (over 90% is stearic acid), are prepared by reversed-phase chromatography [45].

GM1 species completely homogeneous in the ceramide moiety are prepared from the GM1 species containing 18- or 20-sphingosine by specific alkaline hydrolysis to yield lyso-GM1, a GM1 derivative lacking the acyl chain. Acylation of lyso-GM1 with an activated acyl chain or with a reactive anhydride yields completely homogeneous GM1 with defined structure [45].

### 5.3. Preparation of Radioactive GM1 and its Derivatives

There is a long tradition in the use of radioactive gangliosides in studies aimed to understand their metabolic pathway and intracellular trafficking. Several procedures are available for the preparation of radioactive GM1, containing ^3^H or ^14^C in various parts of the molecule [46,47]. GM1 can be ^3^H-labeled at position 3 of sphingosine by oxidation with 2,3-dichloro-5,6-dicyanobenzoquinone, followed by reduction with tritiated sodium borohydride and reversed-phase HPLC to separate the two diastereoisomers. GM1 is isotopically ^3^H-labeled at C6 of the external galactose by oxidation with galactose-oxidase. The aldehyde is reduced with ^3^H-labeled sodium borohydride. GM1 labeled at C11 of sialic acid is obtained by preparing the neuraminic acid-containing GM1, then re-N-acetylated with ^3^H- or ^14^C-labeled acetic anhydride. GM1 labeled at the fatty acid moiety is prepared from lyso-GM1 by acylation with ^3^H- or ^14^C-labeled fatty acid to reconstitute the GM1 structure. Catalytic hydrogenation of the GM1 sphingosine double bond with ^3^H-labeled sodium borohydride allows preparation of GM1 containing ^3^H-labeled sphinganine. A general scheme for the preparation of radioactive GM1 is reported in Figure 2.

Species of ^3^H-labeled, photoactivable GM1 derivatives have been prepared. They contain a nitro-phenyl azide group that yields a highly reactive nitrene derivative under illumination. They have the nitro-phenyl azide at the end of ceramide moiety and tritium in the oligosaccharide chain, or the nitro-phenyl azide linked to the position 6 of external galactose and tritium at position 3 of sphingosine. In addition, a tritium labeled and photoactivable GM1 oligosaccharide has been also prepared [48,49]. Another GM1 derivative carrying a diazirine group and that is 125I-labeled at the ceramide moiety has been described [50]. The structure of radioactive and photoactivable derivatives of GM1 is shown in Figure 3.

GM1 containing fluorescent, paramagnetic, and biotinyl groups have been described and detailed labeling procedures were recently reviewed [51].

## 6. GM1 and Cholera Toxin

The knowledge that gangliosides, together with sphingomyelin, ceramide and cholesterol, are the drawing forces for the dynamic formation of lipid rafts, introduced the concept that GM1 represents a marker for these membrane domains. Cholera toxin is frequently used for the rapid but not necessarily consistent detection of GM1 in the total homogenate, in subcellular fractions or in membrane fractions. In the early 1970s, it was reported that the cholera infection depended by the interaction between the *Vibrio Cholerae* toxin and the mucosal surface [52]. Rapidly, by in vitro approaches, ganglioside GM1 was indicated as the receptor for the toxin [53,54,55], showing an association constant around 10^−9^ M. Both GM1 and its oligosaccharide are bound by cholera toxin [53]. Identification of GM1 in the total homogenate, in subcellular fractions or in membrane fractions by immune-spot with cholera toxin is widely used. Similarly, immune-microscopy is used for recognition of GM1 and of GM1 aggregates on the cell surface, without taking into account the information, too frequently forgotten, that glycosphingolipids are cryptic components of our plasma membranes [56,57,58]. Results should require additional controls, taking also into account that the knowledge on the ability of cholera toxin to recognize different glycans has grown in recent years. Within glycan recognized by the toxin, we can cite IV2αFucGM1, IV3αGalGM1, V3βGalIV4βGalNAcGM1b, Lex trisaccharide and several fucosylated and sialylated glycoproteins. Some of these glycans show affinity constant of the same order of that for GM1. In addition, we need to recall that when applied on a surface, almost all the nervous system gangliosides, depending by their quantity, are recognized by cholera toxin. A detailed commentary on this topic has been reported by Chiricozzi and collaborators in [55].

## 7. The Fate of GM1 Administered to Cells in Culture

Gangliosides were added to the cell medium of cultured cells to increase the cell ganglioside content and follow the activation of cell membrane proteins [59]. GM1 molecules carrying a specific probe, like a radionuclide, a paramagnetic or a photoactivable group [60], and added to the cell culture medium have been used to study the capability of gangliosides to associate to the plasma membranes and their fate once uptaken by the cells. Three different forms of GM1 association have been recognized to occur [60]. The “serum-removable” form is rapidly detached from the cells by washing cells with a medium containing proteins, or albumin solutions. This form of association corresponds to the mild interaction of ganglioside micelles with the cell surface, probably mediated by electrostatic interactions. A second form of association called “trypsin-removable”, corresponds to micelles and a few monomers strongly interacting with the hydrophilic portion of membrane proteins protruding into the cell environment; this is the less representative form of GM1 association. The action of trypsin cuts these membrane protein portions associated to the ganglioside. Then, there is the “trypsin-stable” form of association, corresponding to monomers stably inserted into the plasma membrane (see Figure 4a).

The distribution of these three forms of association is dependent by the experimental conditions: the presence of proteins in the cell culture medium, time of incubation and ganglioside concentration. Proteins in the medium reduce the number of free micelles in solution; a long incubation allows to have more monomers for the trypsin-stable form of association (those that enter into the plasma membranes are no more available for the micelle–monomer equilibrium and new monomers are released from the micelles); the lower the ganglioside concentration the grater the % of monomers in solution (we recall that the monomer concentration cannot exceed 10^−9^ M). Figure 4b shows the distribution of the three association forms of GM1 administered to human fibroblasts cultured in medium without proteins, as a function of incubation time and ganglioside concentration. Administration of radioactive GM1 at a concentration between 10^−6^ M and 10^−7^ M allows to have an additional quantity of membrane GM1 indistinguishable from- and diluted into the endogenous ganglioside, thus useful to trace the ganglioside metabolic pathway and fate of the administered ganglioside [61]. Radioactive GM1, as all the gangliosides, reaches the lysosomes where it is degraded to single sugars, sphingosine and fatty acids. The largest part of sialic acid and sphingosine is exported from the lysosomes recycled for the biosynthesis of sialoglycoconjugates and sphingolipids [62]. A minor portion of GM1 escapes this pathway and is converted into GD1a by a sialylation process [61]. Administration of radioactive GM1 to cultured fibroblasts is one of the simplest and most reliable procedure for diagnosis of sphingolipidosis like GM1 and GM2 gangliosidosis and to follow the recovery of lysosomal enzyme activity following treatments [63].

## 8. Neurotrophic and Neuroprotective Properties of GM1

GM1 is one of the main gangliosides used for biological experiments: it has been administered to a variety of cells in culture. The aim was to increase the GM1 membrane content, to modify the membrane ganglioside pattern and the membrane organization, influencing thus the membrane enzymes and the membrane receptor properties. 

The exogenous addition of GM1 to cultured neurons alters the responses to signals from the surrounding environment, possibly due to specific interactions with membrane proteins. This introduced the statement that gangliosides are associated with brain plasticity [66,67,68].

GM1 modifies the process of differentiation, amplifies responses to neurotrophic factors, protects against excitatory amino acid-related neurotoxicity by limiting the downstream consequences of receptor overstimulation, and reduces acute nerve cell damage by blocking cytotoxicity and potentiating neurotrophic factors [69]. Administration of ganglioside mixtures containing GM1 influenced the recovery processes of both cholinergic and adrenergic nerve fibers in experimental models of peripheral sympathetic regeneration and re-innervation (preganglionic and postganglionic anastomosis) [70,71]. The role of GM1 in induction of neurite sprouting has been elucidated [72,73,74], as well as the role of membrane-associated sialidase Neu3 in producing GM1 from polysialogangliosides. Many of the molecular interactions necessary for the neurotrophic and neuroprotective effects exerted by GM1 require that GM1 is inserted into lipid rafts. In this contest, GM1 replaces or potentiates the actions of neurotrophins in several experimental approaches [75,76,77] by modulating the interaction with their receptors, including the GDNF (glia cell-derived neurotrophic factor) receptor complex [78] and neurotrophin tyrosine kinase family receptors (Trk) [79,80,81,82,83,84,85,86]. The role of GM1 in modulating Trk activity has been studied in detail.

Exogenous GM1 stimulates Trk kinase activity, receptor autophosphorylation, and dimerization in various cell types [79,80,81,82,83,84,85,86]. A significant proportion of Trk receptors in neurons is typically associated with lipid rafts or GM1-enriched membrane domains [87,88,89,90,91,92]. Co-localization of GM1 and Trk receptors in lipid rafts has been proposed to be necessary for TrkA phosphorylation in cultured cells [83], brain tissues [93], and in vivo [86,94]. Glycosylation of Trk is reported to be necessary for targeting of Trk into GM1-lipid rafts [95] and for GM1-derived activation of the receptor [80]. However, we have recently shown that in neuroblastoma cells, TrkA does not belong to lipid rafts where GM1 is located, but its interaction with GM1 and the following stabilization which leads to its autophosphorylation involves only the GM1 oligosaccharide chain and the extracellular portion of TrkA, that may flop down on the plasma membrane approaching the GM1 oligosaccharide chain [96].

Importantly, using PC12 cells typically poor in the GM1 content, it has been shown that the exogenous administration of GM1 strongly enhances nerve growth factor (NGF)-mediated TrkA activation [82,83].

Finally, using neuroblastoma cells deficient in endogenous GM1, it has been demonstrated that NGF did not elicit the autophosphorylation of the Trk protein, but the rescue of GM1 content recovered the responsiveness of Trk to its ligand [97]. This evidence strongly suggests that GM1 is necessary for the normal functioning of Trk protein. 

## 9. The Fate of GM1 Administered In Vivo within the Brain

Of course the possibility of translating the GM1 ganglioside neuroprotective and neurorestorative functions reported in vitro to functional recovery of degenerated nervous system in vivo, relies the capacity of injected ganglioside to reach the brain structures. The brain incorporation of administered gangliosides has been investigated by independent studies [98,99,100,101,102,103]. Although these studies showed that exogenous administered gangliosides are taken up by the nervous system as well as from all other peripheral organs (i.e., liver, muscle, kidney, blood), disagreement are related to the degree of uptake and the real quantity of ganglioside into neuronal cells. A later study by Tettamanti and co-workers clarified how GM1, intravenous administered to rats, is taken up by the brain and binds to the capillary network penetrating into the neuronal cells [62]. Within the cells, GM1 is associated both to plasma membranes and to intracellular structure, undergoing metabolic processes by the formation of a number of products of both catabolic and biosynthetic origin [62]. The latter finding suggests and highlights that the quantity of GM1 available to sustain its biological activity through the interaction with plasma membrane receptor is time-limited and dependent on the rate of its catabolic recycling. However, the ganglioside distribution that has been observed in the rodent models has not been successful translated in humans [7,104] as often, the two species display different blood–brain barrier (BBB) permeability rates to different molecules. Accordingly, to obtain a relevant result in Parkinson’s patients, the quantity of injected GM1 was remarkable, and even for Alzheimer’s patients, GM1 had to resort to intracranial injections [105], an administration route not compatible with human life. As reported below, many unsuccessful efforts have been made by researchers to find soluble analogues of GM1 [106,107,108] or to increase the amount of GM1 enzymatically [109]. However, new light comes from our recent recognition that the soluble oligosaccharide of GM1 represents the bioactive portion of the entire ganglioside [96,110,111,112,113]. As explained in detail below, these data establish a reasonable basis for considering the GM1 oligosaccharide as an agent that exceeds the GM1 pharmacological limits and could show significant therapeutic benefits for neurodegenerative diseases of the central nervous system. 

## 10. GM1 as a Therapeutic Drug

The use of drugs containing the pure ganglioside mixture from calf brains started in 1976 in Italy with the name of Cronassial and in the following years, in many European countries, in South and Central America, in Asia and Africa. The drug was approved for treatment of peripheral neuropathies [114,115,116,117]. Later, in 1985, the ganglioside GM1 entered in the market with the name Sygen, prescribed for neurodegenerative diseases and possibly for cerebral and spinal cord injuries [118]. Clinical trials for the treatments of cerebral ischemia and dementia, stroke and spinal cord injury were developed with ganglioside GM1 [7,119]. Unluckily, at the beginning of the 1990s, part of the medical society claimed that gangliosides circulating in the blood stream stimulated the production of anti-ganglioside antibodies being responsible for the occurrence of Guillain-Barré syndrome (GBS) [120]. Following this, in many countries, the drug was withdrawn at the end of 1993 and the trials could not be completed.

Gangliosides are not immunogenic compounds, but on the other hand, they become immunogenic when carried by an adjuvant, such as a microorganism. Microorganisms or viruses like Campylobacter jejuni, Mycoplasma pneumonia, and cytomegalovirus that have glycans or partial glycans at the surface which are a copy of the ganglioside head groups are immunogenetic when infecting human body. The immune system responds then producing antibodies that recognize gangliosides [121].

The claim that pure gangliosides are not immunogenic when injected in patients derives from the extensive use of them in recent decades [120]. In this period, over 16 million patients received the total ganglioside mixture or GM1 occasionally; five Alzheimer’s disease patients received ganglioside GM1 in continuous into the brain lateral ventricles for one year [105,122,123]; over one hundred Parkinson’s disease patients received GM1 daily subcutaneously, for two years [124,125]; over 700 patients with spinal cord injury received GM1 daily for up to 5 years [126]. No one of the controlled daily treated patients developed serum antibodies nor GBS [7,127]. Indeed, the incidence of developing serum antibodies or neurological autoimmune syndrome in patients treated sporadically with gangliosides was overlapping to the number of case referring to people that never received gangliosides. Today, the notion that the administration of gangliosides is not associated to the onset of GBS, seems to be sounder, and in recent years, clinical trials at different stages, still involve ganglioside GM1. On the other hand, the use of gangliosides extracted from tissues or produced in part or at all by bacteria requires great care to avoid any type of contamination that could add immunogenic properties to the drug. For detailed information on the non-association of gangliosides to GBS, please refer to Sonnino et al. [120]. In the following sections, we provide information about the neurodegenerative diseases treated with GM1.

### 10.1. GM1 and Parkinson’s Disease

In total, 5%–10% of Parkinson’s disease cases are of known genetic origin, whereas the large majority, termed sporadic, has only age as the major risk factor [106,128]. Unlike familial forms, both genetic and environmental factors have been reported to play a synergistic role in the pathogenesis of sporadic Parkinson’s disease, but although various theories have been suggested, none have successfully explained the accumulated data regarding the diverse central and peripheral manifestations of sporadic Parkinson’s disease [106,128]. The accumulation and aggregation of α-synuclein is generally considered to have a central role in Parkinson’s disease.

Recently, a theory emerged that defines a central role for ganglioside GM1 whose levels, along with its catabolic precursor GD1a, diminish with aging and/or under epigenetic influences [9,106]. It was proposed by Lars Svennerholm† (Department of Clinical Neuroscience, Göteburg University, Sweden), one of the scientific father of the research on the human brain gangliosides, that brain GM1 level varies significantly among individual of same age and that it declines progressively with age [10,108]. In some subjects, presumably those starting life with lower levels, GM1 may reach a point in later years when it falls below the threshold level necessary to maintain dopaminergic and forebrain neurons viability, thereby gradually leading to sporadic Parkinson’s disease [9,106,108]. As the population ages and GM1 continues to decline, the number and percentage of persons with this condition will multiply [106].

In Parkinson’s disease patients, a significantly deficient expression in genes involved in ganglioside synthesis (*B3GALT4*, *ST3GAL2*) was reported and GM1 deficiency in the *substantia nigra* [107], in the occipital cortex [129] and in various peripheral tissues [106] was observed. These findings suggest a ganglioside systemic deficiency that would correlate with the systemic symptoms of sporadic Parkinson’s disease. The decreasing expression of mentioned genes and the resulting decrease in brain of a-series gangliosides (GM1, GD1a) presume impaired composition and disorganization of neuronal plasma membrane, which may increase the neurons vulnerability to degeneration [130].

GM1 is one of the predominant brain gangliosides with demonstrated anti-neurotoxic, neuroprotective and neurotrophic actions in vitro and in vivo [2,6,7,104,131]. Accordingly, GM1 acts through diverse mechanisms inhibiting inflammation, excitotoxicity and oxidative stress reactions, modulating neurotrophic factor signaling, membrane integrity, calcium homeostasis and α-synuclein aggregation. As elaborated in recent reviews, these neuronal functions can become gradually compromised as GM1 levels recede with aging and/or under epigenetic influences [2,6,7,9,104,106,131]. In this scenario, the possibility of GM1 as initiator of Parkinson’s disease may particularly make sense: if GM1 level is reduced, the GM1 mediated mechanisms may be compromised leading progressively to the complex cascade of interrelated events that cooperates to determine Parkinson’s neurodegeneration.

Among other, two functions have commanded special attention regarding Parkinson’s disease: neuroprotective signaling via neurotrophic receptors and α-synuclein interaction. It has been hypothesized that the reduced level of plasma membrane GM1 in Parkinon’s neurons can trigger the neurodegenerative process by a failure in neurotrophic signaling (e.g., BDNF/GDNF/NGF) together with a reduction of clearance promoting the α-synuclein accumulation [7,9,104,106,129,130]. Along with this, recent studies strongly demonstrated the specific binding of tetrameric α-synuclein to GM1 which promotes its α-helical conformation against the beta-sheet-rich state of α-synuclein that has been associated with its pathological aggregation in Parkinson’s disease [132,133,134]. Additionally, Schneider recently reported that GM1 modifies the alpha-synuclein toxicity and is neuroprotective in a rat α -synuclein model of Parkinson’s disease reducing the size of α -synuclein positive aggregates [123].

From the above research, a GM1 replacement therapy was developed. Several trials were developed starting from the 1990s. Schneider and co-workers carried out a study (ClinicalTrials.gov NCT00037830) with groups of patients up to one hundred daily treated with 100–200 mg of GM1 for up to 5 years [124]. GM1 was administered intravenously or subcutaneously. The treatment showed some positive effects exerted by GM1, such as a partial restoration of dopamine transporter functional level in the striatum, motor symptoms improvement and lowering the disease symptom progression [124,125,135]. This suggested that GM1 may have symptomatic and potentially disease modifying effects on Parkinson’s disease. However, the groups were still small and the results were not clearly sufficient to imagine GM1 as an official authorized drug for Parkinson’s disease. As stated above, a too small quantity of administered GM1 cross the BBB and reaches the damaged neurons. It is believed that the positive effects exerted by GM1 on the Parkinson’s disease patients is mainly due to the passage of a minor part of administered GM1 through the damaged-inflamed area of the BBB, present in patients with advanced pathology [136,137,138]. At this stage, too many neurons are lost and the protective and neurotrophic effect cannot be exerted by GM1 due to a lack of target cells. On the other hand, in the first stage of the pathology, when the BBB is not yet damaged, it is the BBB itself that prevents GM1 from fulfilling its properties, hampering it from meeting neurons [139,140].

### 10.2. GM1 and Alzheimer’s Disease

Several papers reported on the possible negative role of GM1 in the formation of neurotoxic β-amyloid fibrils at the cell surface [141,142,143,144,145]. A claim was made that GM1 would act favoring the deposition of the aggregates following specific interaction requiring the GM1 high clustering [146,147,148].

The membrane complex lipids are very dynamic and according to their physico-chemical properties and their proportion in the membrane determine the membrane organization. Any change of one of the membrane components activates a dynamic process that modifies the membrane lipid distribution and the composition of the lipid rafts [9,96,149,150,151,152,153].

As reported in an above paragraph, there is a loss of gangliosides in the brains or at least is some areas, from Alzheimer’s disease patients [154]. However, looking at a specific brain area, it was found that in the AD frontal cortex, where it seems that the disease starts, the GM1 content in lipid rafts doubles that of aged matched healthy controls [155]. Therefore, the question was: is it possible to report a normal membrane organization by reporting the correct ganglioside content into the the brain? To answer this question, five Alzheimer’s disease patients aged 54 to 70 received a continuous treatment with GM1. Patients received up to 30 mg/day of GM1 ganglioside by continuous injection into the brain frontal lateral ventricles for 12 months. These patients became more active, had improved reading comprehension, were able to perform activities such as writing reports and short letters on a computer, and showed increased appetite and sexual desire [105,122,123]. They displayed increased production of the catabolic products of dopamine and serotonin as proof of an improvement of the transmitter functions. In particular, they showed a strong increase of the level of homovanillic acid in the cerebrospinal fluid, this supporting a restoration of dopaminergic neurons functions. On the contrary, in a trial in which GM1 was intramuscular administered, no amelioration was observed [105,156]. These two results are not in contrast. As reported, the administered GM1 that enters into the blood stream does not cross the BBB or cross it in a very scant quantity, surely not comparable with that available in the brain when administered through the ventricles.

### 10.3. GM1 and Spinal Cord Injury

Several trials were carried out on the use of GM1 to treat spinal cord injury [157,158,159,160,161]. Probably the bigger group was one of 760 with acute and chronic spinal cord injury, chosen from a bigger group of 3130 from 28 centers [126]. Patients were divided into two groups and treated with 100 or 200 mg of GM1 daily administered intravenously. Treatment began 72 h following injury and lasted 8 weeks. At the end of treatment, the proportion of patients with marked recovery was significantly greater in both GM1 dose groups as compared to placebo. However, no differences could be found at week 26. The conclusion was that the GM1 treatment allows a significantly faster, although ultimately not greater recovery. Moreover, other trials found similar results, confirming an acceleration of the recovery due to GM1 treatment [162,163,164].

### 10.4. GM1 and Peripheral Neuroprotection by Oxaliplatin Treatment

Oxaliplatin is a very powerful drug against gastrointestinal tumors; however, it is characterized by severe peripheral neurotoxicity. Sixty patients with gastrointestinal tumors were daily injected with 100 mg of GM1 for the 3 days following the chemotherapy treatment. The treatment showed some protection against the oxaliplatin neurotoxicity. 

As reported by recent reviews [165,166], GM1 has been studied as a possible treatment in brain damage resulting from hypoxic phenomena mainly in two contexts: hypoxic damage to white matter and ischemia brain damage. The first occurs with increasing frequency in premature infants, with consequent neurological and cognitive *sequelae* that limit the quality of life and a possible etiology has been found in the susceptibility of not completely mature oligodendrocyte precursors to hypoxia [167]. On the other hand, ischemic damage is increasingly common in aging adults and is mainly caused by thrombotic obstruction of cerebral vessels, while only a minority have a hemorrhagic origin [168].

### 10.5. GM1 in Brain Damage of Premature Infants

Recently, a small clinical study was conducted in China on a series of 76 premature infants where the treatment with GM1 administered intravenously for 14 days resulted in a significant reduction in the damage to white matter around the cerebral ventricles after 7 and 14 days after birth compared to controls subjected to the standards of care [169]. At the same time points, there was a stronger reduction in serum levels of IL-6, neuronal specific enolase NSE and S100β in the group of children receiving GM1 compared to controls. After 1 year, the group of children treated with ganglioside had an IQ higher than the control group in the aspects of social adaptation, gross motor, fine motor, language and personal social contact. Importantly, no side effects were reported in GM1 receiving infants [169].

Moreover, severe hypoxic-ischemic encephalopathy could affect perinatal infants with not yet clarified reasons [170]. The co-treatment of GM1 with human recombinant erythropoietin has been shown to significantly improve short-term effects and long term neurological symptoms [171].

### 10.6. GM1 in Ischemic Damage of Adults

Given its good neuroprotective effects in animal experiments [165,172,173,174,175,176,177,178,179], GM1 successfully attracted the attention of neurologists and began to be used in patients with acute ischemic stroke; however, its clinical efficacy still needs to be demonstrated. A systematic review of 12 controlled clinical trials involving about 2300 ischemic patients was published by Candelise and Ciccone in the Cochrane Stroke Group trials register [119]. The results did not show any significant differences with respect to the incidence of disability and fatality between GM1 treated and not treated groups. The authors conclude that caution should be exercised in the use of GM1 in the clinic, mainly due to the supposed association with Guillan-Barré syndrome even if a cause-effect relationship has not yet been demonstrated. In fact, GM1 continues to be used in Asian countries in numerous pathologies without considerable adverse effects reported.

Reasons for this translational failure are unknown and different issues may have interfered with therapies, including inadequate dosing and study design flaws [180].

In the four main multicenter clinical trials [181,182,183,184], although there was no significant difference in the mortality rate between the groups of ischemic patients treated with GM1 and the placebo-receiving controls, the subjects treated with GM1 presented an improvement trend in neurological scores (as reviewed by Magistretti and collaborators [7]).

Interestingly, in animals, GM1 administration continued to give positive results in ischemia/reperfusion models. In these models, recently, Choucry and co-workers [185] observed that 30 mg/kg of GM1 administered immediately after occlusion of the carotid arteries and then 3 h after reperfusion and for following 6 days led to a clear improvement in the cognitive functions of rats (recovery of the deficit in spatial memory retention and acquisition induced by ischemic damage). This effect was associated with a modulation of TrK / p75NTR / NGF signaling in favor of neuronal survival compared to apoptosis.

## 11. Administration of GM1 and Modified GM1 to *B4galnt1^−/+^* Mice

A functional connection between human Parkinson’s disease pathogenesis and GM1 was confirmed in an experimental animal model.

The *B4galnt1* gene codes for the enzyme N-acetylgalactosamine transferase, necessary to add a residue of N-acetyl-galactosamine to the ganglioside GM3 to yield ganglioside GM2. GM2 is the precursor of more complex gangliosides of a and b series. In the absence of GM2, the complex gangliosides are not synthesized and all GM3 is shifted to the synthesis of gangliosides GD3. 

The total ablation of the *B4galnt1* gene in mice (*B4galnt1^−/−^* mice), leads to the complete absence of gangliosides of the ganglio-series and increased GM3 and GD3, accompanied by a phenotype of severe neurodegeneration that recapitulates severe parkinsonism [186].

Instead, the heterozygous mice (*B4galnt1^+/−^* mice), are characterized by a partial reduction of ganglioside GM1 and GD1a (GD1a is converted to GM1 by the membrane associated sialidase Neu3) with respect to the wild-type mice and present a mild manifestations of Parkinson’s disease better recapitulating the sporadic and progressive form of the disease [106,107,187,188,189]: α-synuclein elevation and aggregation within central and peripheral neuropathological lesions, nigro-striatal degeneration and worsening of motor dysfunction.

Both the homozygous and heterozygous mice have been subjected to GM1 replacement therapy. In addition, the efficacy of a semisynthetic GM1 analogue known as LIGA20, in which the fatty acid linked to sphingosine was substituted by the dichloroacetyl group, has been also investigated. In the latter case, the presence of two chlorine atoms increases the membrane permeability to the molecule.

Intraperitoneal daily treatment with GM1 gave statistically not reliable recovery of the Parkinson’s Disease symptomatology [107]. This negative result is probably due to the very low quantity of GM1 that crosses the BBB reaching the neurons, to finally express its neuroprotective function. On the other hand, LIGA20 is able to cross the BBB more efficiently than GM1 because of the chlorine content in the lipid moiety. Thus, systemically injecting LIGA20 into affected mice significantly attenuated the parkinsonism symptoms restoring the expression of tyrosine hydroxylase in the *substantia nigra* and dopamine release in striatum, reducing the α-synuclein deposition and aggregation, and recovering motor impairment [107]. In other experimental models of Parkinson’s disease, GM1 has been shown to be successful when injected directly into the cerebral ventricles. Overall, these results indicate that when the ganglioside reaches the neuronal membranes, it is able to reduce and counteract the progression of the disease [7,130].

With regards to a translational aspect on humans, a therapy based on continuous and repeated intracranial injections of GM1 would be extremely invasive and difficult to achieve. Moreover, the membrane permeable derivatives of GM1, like LIGA20 have been shown to be toxic compounds when administered for long time. A further problem to consider concerns the time of action of the administered GM1. In fact, endogenous gangliosides are employed into a complex metabolic process involving biosynthesis, intracellular trafficking and catabolism. In neurons, the ganglioside membrane turnover is rapid. We know from many experiments that GM1 administered to cells is rapidly taken up by the cells and directed to the lysosomes where it is catabolized [62]. Thus, to maintain the effect exerted by exogenous GM1 on neurons, it should be administered to Parkinson’s disease patients several times a day.

## 12. The Oligosaccharide Chain is the Functional Portion of GM1

Although the effects of GM1 are quite evident in vitro, in animal models as well as in humans, the molecular mechanism through which GM1 may induce neurodifferentiation and exert neuroprotection and, when deficient, neurodegeneration, should be addressed.

GM1 is known to modulate several cell processes through interactions occurring at the cell surface [2,6,26,131]. GM1 is an amphiphilic compound, with the lipid and the head group moieties imbibed into two different environments. Modulation of surface proteins’ function can be due to hydrophilic as well as to hydrophobic interactions. In addition, changes of the GM1 concentration or segregation, are necessarily followed by general membrane reorganization and this could force other membrane components to interact and finally to modify the proteins’ activity [186,190,191].

The TrkA receptor has been studied in detail in different in vitro cell models, and its activation has been proved to require the interaction with GM1 and the establishment of the three-component complex TrkA-NGF-GM1 [96]. Despite this evidence, the way through which the interaction could occur was missing. On one hand it is already known that GM1 is a major component of lipid rafts and that the ganglioside structures are the driving forces necessary to form them. On the other hand, TrkA does not normally reside within lipid rafts since was shown to not co-localize in GM1 enriched domains [96]. Thus, the question arises about the mechanism by which GM1 and TrkA could approach each other in order to regulate the cascade of events sustaining the neuronal cell functions. Is it an interaction involving purely the hydrophobic component of GM1, the ceramide, or specifically the hydrophilic oligosaccharide portion, or are both portions of the ganglioside necessarily required to interact with the receptor?

In 1988, a pioneering paper by Shengrund and Prouty observed that the GM1 oligosaccharide chain promoted neuritogenesis [192].

In 2012, Ledeen identified the membrane brain permeable analog of GM1, LIGA20, as an effective (albeit toxic in the long-term) alternative to GM1 [106,107]. Treatment of parkinsonian *B4galnt1* GM1-deficient mice, with LIGA20 induced beneficial effects, including the reduction of *substantia nigra* alpha-synuclein aggregates [106,107]. The most important finding of this study was the identification of a hydrophilic GM1 derivative, modified on the ceramide moiety with a dichloroacetyl group instead of the acyl chain, but keeping the entire oligosaccharide intact, which still maintained trophic potential. This suggested that the ceramide structure is not critically related to GM1 modulatory effects.

In 2015, Scheneider noted that the plasma membrane GM1 increase by intraventricular injection of *Vibrio Cholerae* sialidase exerted a neuroprotective effect on the damaged nigrostriatal dopaminergic system of MPTP mice [109]. This enzyme removes sialic acid residues from brain polysialogangliosides (i.e., GD1a), thus increasing plasma membrane GM1, further evidence that the oligosaccharide may act as the mediator of GM1 function [109].

From all these premises emerged our idea that the sole oligosaccharide portion of the GM1, (β-Gal-(1-3)-β-GalNAc-(1-4)-[α-Neu5Ac-(2-3)-]β-Gal-(1-4)-Glc, II^3^Neu5Ac-Gg_4_, OligoGM1), represents the bioactive portion of GM1 that protruding into the extracellular environment acts at the cell surface by interacting with plasma membrane proteins 

### 12.1. Administration of OligoGM1 to Cells in Vitro

We recently reported that the soluble GM1 oligosaccharide replicates the neurotrophic and neuroprotective properties of the entire GM1 molecule [96,110,111]. Using murine neuroblastoma Neuro2a (N2a) cells, we demonstrated that OligoGM1, without entering the cell, directly interacts and stabilizes the TrkA-NGF complex, leading to the activation of the ERK1/2 downstreaming pathway. This event in turn triggers cell differentiation and MPTP protection, similarly to that induced by the whole GM1. TrkA inhibition blocks both OligoGM1 induced differentiation and protection, indicating that the OligoGM1-TrkA interaction is the trigger for GM1-mediated functions.

Proteomic analysis revealed that OligoGM1-treated N2a cells expressed over 300 proteins absent in control cells. Importantly, these proteins are mainly involved in biochemical mechanisms related to neuroprotection.

Using primary cerebellar granule neurons obtained from mice, the neurotrophic properties of OligoGM1 were further proved [193]. Accordingly, OligoGM1 enhances neuron clustering, arborization and networking by higher phosphorylation rate of FAK and Src proteins, the intracellular key regulators of neuronal motility. In parallel, OligoGM1 receiving cells express increased level of specific neuronal markers (β3-Tubulin, Tau, Neuroglycan C, Synapsin) anticipate the expression of complex ganglioside and reduce the level of simplest ones, suggesting an advanced stage of maturation compared to controls. Concerning its mechanism of action, as for N2a cells, OligoGM1 interacts with neuronal surface without entering the cells, and induces TrkA-MAPK activation as an early event underlying its neuronal effects.

These results highlight a GM1-specific role in neuronal differentiation and protection determined by its oligosaccharide chain, which interacts with plasma membrane neurotrophin receptors and triggers the activation of intracellular pathways responsible for neuronal functions. 

### 12.2. OligoGM1 and TrkA Interaction Study in Silico

The interplay between TrkA receptor and GM1 ganglioside has been reported in several papers [79,82,83,85,194,195], where it was emphasized that the NGF receptor requires the presence of GM1-enriched membrane to be active, while the GM1 absence negatively correlates with TrkA function. 

In 2015, Fantini reported on the presence of a GM1-binding domain in the extracellular domain of Trk receptor, suggesting that GM1 oligosaccharide could act as an endogenous activator of Trk receptor [196]. Recently we added more details on the molecular basis of the GM1-TrkA interaction. The crystallized structure of TrkA-NGF complex is resolved as a dimer and presents a pocket in the interface of receptor-ligand interaction. By in silico molecular docking studies we found that the OligoGM1 perfectly occupy this space and concomitantly reduces the free energy associated to the TrkA-NGF complex from approx. −7 kcal/mol to approx. −12 [110]. This finding indicates that GM1 oligosaccharide stabilizes TrkA-NGF association and suggests a specific molecular recognition process between the GM1 sugar code and the extracellular domain of TrkA.

### 12.3. Administration of OligoGM1 to B4galnt1^−/+^ Mice

To overcome pharmacological limitations of GM1, the ability of OligoGM1 to reach the central nervous system was verified [112].

Following the intraperitoneal injection of radiolabeled [^3^H]OligoGM1 in adult mice, the radioactivity counts were found to be associated with all brain areas, including the *substantia nigra*. Importantly, OligoGM1 retained its metabolic stability when was extracted from brain tissues as verify by chromatography.

The therapeutic potential of OligoGM1 was then investigated in the *B4galnt4* heterozygous Parkinson’s disease model previously used to demonstrate GM1 efficacy [112]. 

OligoGM1 treatment of these mice (20 mg/kg, intraperitoneally, daily for 28 days) completely rescued physical symptoms, reduced nigral alpha-synuclein aggregates, restored nigral tyrosine hydroxylase expression and striatal neurotransmitter levels, equalizing the wild type healthy condition. These results propose the OligoGM1 as a potential therapeutic candidate that modifies the progression of Parkinson’s disease. Considering the data obtained in vitro it is reasonable to think that the OligoGM1 action is based on the interaction and modulation of membrane protein receptors, once the neuronal target has been reached.

Overall, these recent data confirm that the specific role of GM1 in neuronal function, described in the past, is determined by its oligosaccharide portion which, by interacting with plasma membrane proteins, triggers the activation of intracellular biochemical pathways responsible for neuronal differentiation and protection. It is important to recognize the greater therapeutic translationality of the hydrophilic chain of the GM1, which alone maintains the beneficial neurotrophic properties, but loses the amphiphilicity of the ganglioside, gaining thus the ability to efficiently access the central nervous system (Figure 5).

Thanks to extensive research into the physiological function and pathological implications of ganglioside GM1 [2,6,7,9,104,106,131], we may have a new key to understand the molecular mechanism underlying Parkinson’s disease pathogenesis and recovery. Our new remarkable findings [96,110,111,112,193] call the attention to GM1 oligosaccharide as a completely novel and promising neurotrophic player.

## Figures and Tables

**Figure 1 ijms-21-00868-f001:**
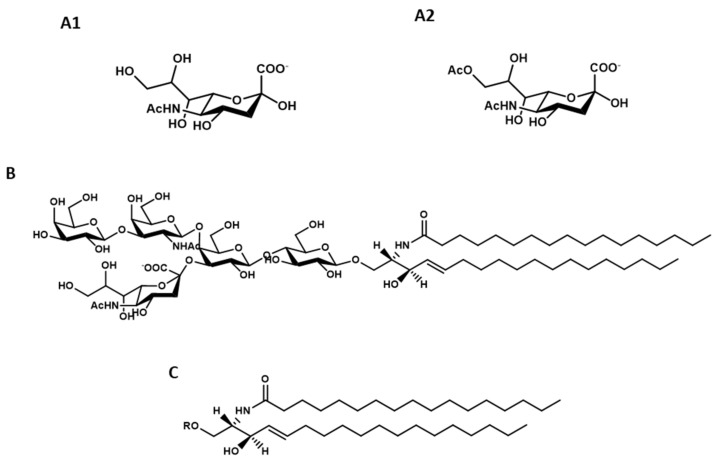
Structure of: *N*-acetylneuraminic acid, Neu5Ac (**A1**), *N*-acetyl-9-acetylneuraminic acid, Neu5,9Ac_2_ (**A2**), ganglioside GM1, II^3^Neu5AcGg_4_Cer (**B**), ceramide, Cer R = oligosaccharide chain (**C**).

**Figure 2 ijms-21-00868-f002:**
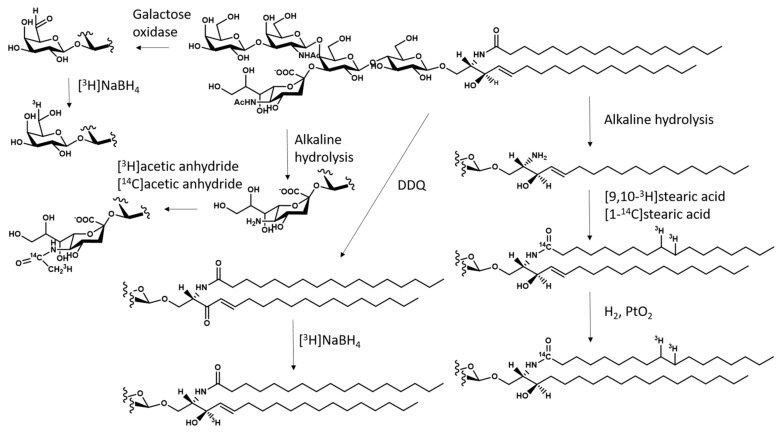
Scheme for the preparation of ganglioside GM1 containing [^3^H] and [^14^C] isotopically inserted into the lipid and sugar moieties (DDQ, 2,3-dichloro-5,6-dicyanobenzoquinone; PtO_2_, platinum oxide; NaBH_4_, sodium borohydride).

**Figure 3 ijms-21-00868-f003:**
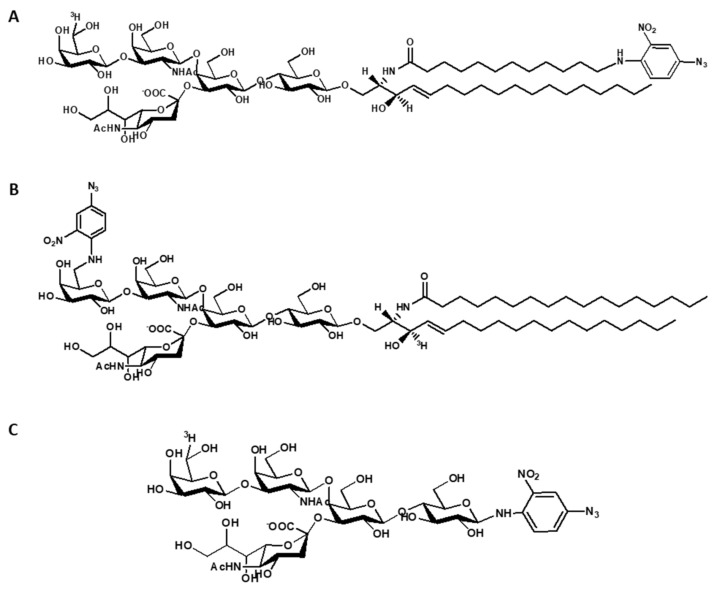
Structure of radioactive and photoactivable derivatives of GM1. (**A**) Structure of [6-^3^H*(IV-Gal)*]GM1(Cer-N_3_); (**B**) structure of 6-N_3_*(IV-Gal)*GM1-[Sph-3-^3^H]; (**C**) structure of [6-^3^H*(IV-Gal)*]OligoGM1(Glc-N_3_).

**Figure 4 ijms-21-00868-f004:**
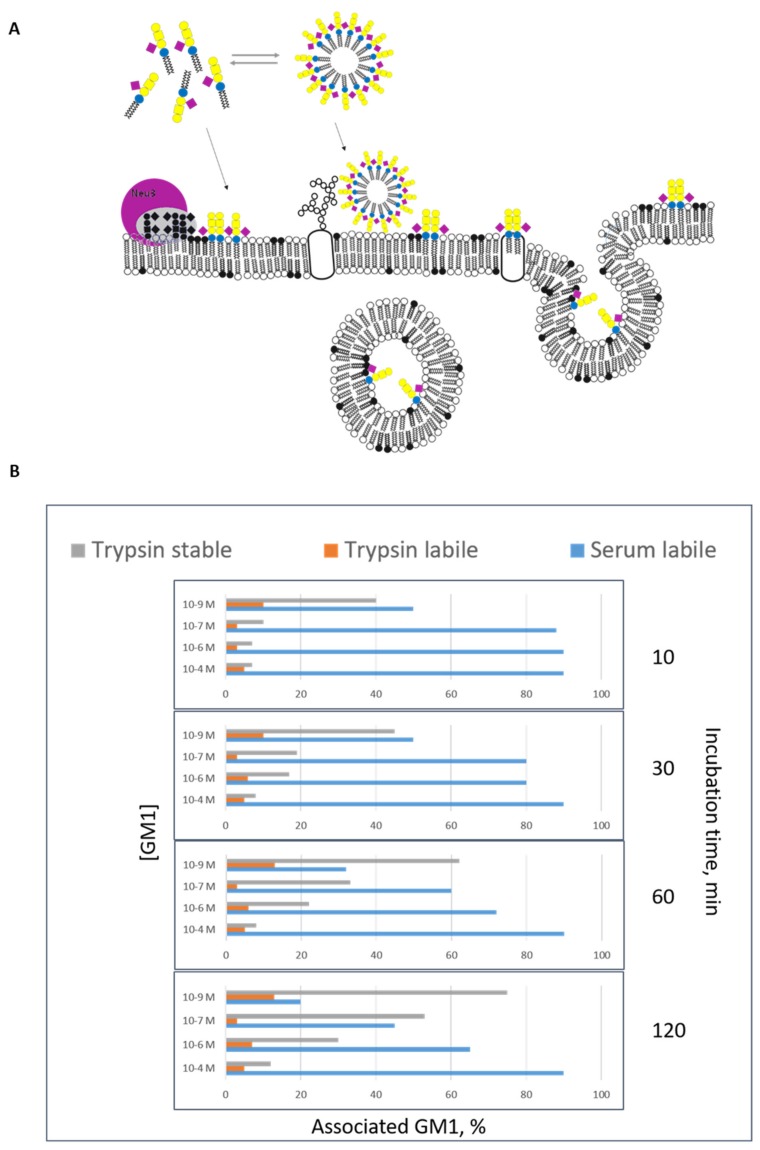
Association of exogenous administered GM1 to cell culture. Ganglioside occurs in water solution in two physical forms, as monomers and micelle. Below a certain concentration, called the critical micelle concentration, gangliosides are found as monomers, whereas above this concentration both micellar and monomer forms are found in equilibrium to each other. The critical micellar concentration for ganglioside GM1 is 10^−9^ M. GM1 interacts with the cell in 3 different forms, serum labile, trypsin labile and trypsin stabile form [63,64]. (**A**) Scheme of the association of ganglioside GM1 administered to cells in culture. Micelles interact with the surface and can be removed by serum washing (serum labile form); the form of cell-associated GM1, the trypsin stabile form, corresponds to GM1 single molecules, which is probably present in the plasma membrane external layer and can be removed by trypsin treatment (trypsin labile form of association). Finally monomers are taken up by the cells entering into the cells (trypsin stable form of association). GM1 sugar code is according to Varki et al. [65]. (**B**) Percent distribution of the three form of GM1 association to rat cerebellar granule cells in culture, as a function of molarity and time [63,64]. As seen in the graph, the proportion of the associated GM1 depends on the incubation time and on the ganglioside concentration of the initial treatment. Decreasing the GM1 concentration, we increase the quantity of GM1 present as monomer instead of GM1 present as micelle in solution and inserted into the cells. Increasing the incubation time, the quantity of GM1 inserted into the cells increases in percentage terms.

**Figure 5 ijms-21-00868-f005:**
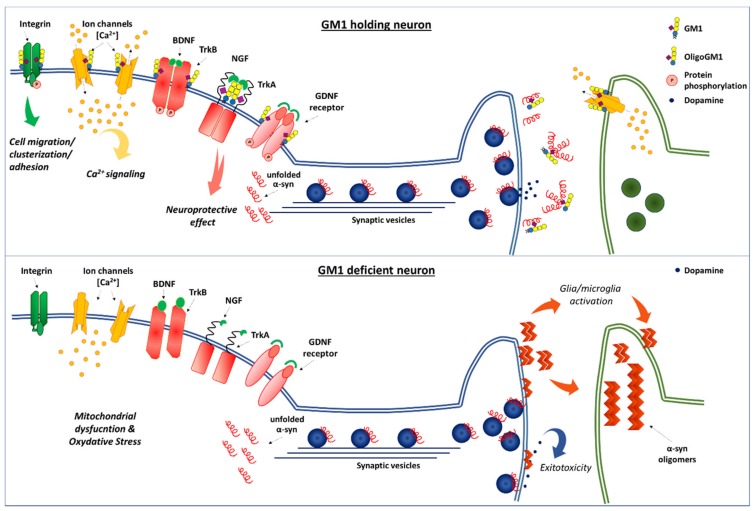
Neuronal function dependent by interaction between the GM1 oligosaccharide and proteins [106]. On the top of the image there is a healthy neuron holding with correct GM1 level that sustains neuron homeostasis and also the clearance of α-synuclein. Some of the fundamental neuronal processes that require the association between GM1 oligosaccharide and protein at the plasma membrane level are represented. The viability of dopaminergic neurons depends particularly on the neurotrophic and neuroprotective signaling throughout Trk(s) and RET receptors [2,9]. Fundamental for neuronal survival is also the GM1 collaboration with ion channels (calcium) and integrin receptors [9]. Finally, it is shown also the α-synuclein association with synaptic vesicles and its release in presynaptic region functioning as a regulator of dopamine neurotransmission [197,198,199,200]. Here the correct level of plasma membrane GM1 maintains, in out hypothesis, the α-synuclein in the non-aggregating forms. On the bottom, a neuron with GM1 deficiency and hence a deficiency of its oligosaccharide, that leads to the loss of all the important neurotrophic signals and to α-synuclein aggregation and accumulation. GM1 sugar code is according to Varki et al. [65].

**Table 1 ijms-21-00868-t001:** Structure of the main brain gangliosides belonging to the ganglioside series 4.

Trivial Accepted/Used Name	IUPAC-IUB Nomenclature	Chemical Structure
AGM1 (to discourage)	Gg_4_Cer	β-Gal-(1-3)-β-GalNAc-(1-4)-β-Gal-(1-4)-β-Glc-(1-1)-Cer
GM1a	II^3^Neu5Ac-Gg_4_Cer	β-Gal-(1-3)-β-GalNAc-(1-4)-[α-Neu5Ac-(2-3)-]β-Gal-(1-4)-β-Glc-(1-1)-Cer
GD1a	II^3^Neu5Ac,IV^3^Neu5Ac-Gg_4_Cer	α-Neu5Ac-(2-3)-β-Gal-(1-3)-β-GalNAc-(1-4)-[α-Neu5Ac-(2-3)-]β-Gal-(1-4)-β-Glc-(1-1)-Cer
GD1b	II^3^Neu5Ac_2_-Gg_4_Cer	β-Gal-(1-3)-β-GalNAc-(1-4)-[α-Neu5Ac-(2-8)-α-Neu5Ac-(2-3)-]β-Gal-(1-4)-β-Glc-(1-1)-Cer
GT1b	II^3^Neu5Ac_2_,IV^3^Neu5Ac-Gg_4_Cer	α-Neu5Ac-(2-3)-β-Gal-(1-3)-β-GalNAc-(1-4)-[α-Neu5Ac-α-Neu5Ac-(2-3)-]β-Gal-(1-4)-β-Glc-(1-1)-Cer
GQ1b	II^3^Neu5Ac_2_,IV^3^Neu5Ac_2_-Gg_4_Cer	α-Neu5Ac-(2-8)-α-Neu5Ac-(2-3)-β-Gal-(1-3)-β-GalNAc-(1-4)-[α-Neu5Ac-α-Neu5Ac-(2-3)-]β-Gal-(1-4)-β-Glc-(1-1)-Cer

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
