# Peer review of "GM1 Ganglioside Is A Key Factor in Maintaining the Mammalian Neuronal Functions Avoiding Neurodegeneration"

_ijms, 2020, doi:10.3390/ijms21030868_

Round 1

Reviewer 1 Report

In this paper named “GM1 ganglioside is a key factor in maintaining the mammalian neuronal functions avoiding neurodegeneration.” the authors mentioned many species of ganglioside GM1 in detailed and different work in the neuronal field focusing to understand its binding properties and its neurotrophic and neuroprotective role. GM1 is one of the main actors in determining the neuronal functions as the progressive reduction of its biosynthesis is being considered as one of the reasons for the neurodegenerative diseases. This manuscript is well written. In my opinion, few minor changes will improve this manuscript more.

Figure 1, it showed a, b and c but it is empty. It would be a formatting problem. The introduction can be more informative and expanded discussing different aspects of the discoveries and other breakthroughs precisely. Figure 4a, the figure caption required more information. You have used several shapes in that figure but it would be helpful if you explain them. Same for the figure 4b.

I am wondering whether the authors generated these figures otherwise they can provide the references at the end of each figure caption.    

Reviewer 2 Report

The manuscript dealing with the knowledge on ganglioside GM1 and its potential against neurodegeneration provides a comprehensive description beginning from its structure, preparations and analysis through its fate in cells to its use in the treatment of neurodegenerative diseases. The presented herein data come from physical analysis and in silicon, in vitro, in vivo and clinical studies. However, some issues, as listed below, should be addressed by the authors.

Major comments

Figure 1 shows nothing. Lines 221-222 since the authors point to GM1 containing fluorescent, paramagnetic, and biotinyl groups their use in relevant studies of neuron functions and their preparation should also be presented. Figure 4a should be labeled and provided with a legend. Moreover, its caption should contain a more detailed description of the processes presented, including all elements presented. Similarly, the caption of Figure 5 should contain a more detailed description of the presented mechanism. Since the paper deals with the use of GM1 in the treatment of neurodegenerative diseases and its inefficient ability to cross the BBB is indicated as a key limitation of this use, it is recommended to provide an additional paragraph presenting the current state of knowledge about the distribution of GM1 to the brain and approaches developed to overcome it. Lines 417-418, please specify a form of α-synuclein (misfolded, oligomers, monomer?) targeted by the specific binding of GM1 resulting in the inhibition of α-synuclein aggregation. Lines 446 and 510, please clarify the control groups in the description of trials. Lines 345-346, precise information about the drug “approved for treatment of peripheral neuropathies” should be provided. Lines 452-456 where were dopamine metabolites measured? It would be valuable to provide IDs of clinical trials referenced in this manuscript. Why are experimental animal models of a functional connection only between Parkinson’s disease and GM1 presented? The main text lacks references in many places, such as in lines 393, 401, 482, 492, etc.

Minor comments:

Line 382 it should be rather the accumulation than the elevation of α-synuclein.

Round 2

Reviewer 2 Report

Figure 4a should include the legend for the colorful graph elements presented there.
